# Severity Patterns in COVID-19 Hospitalised Patients in Spain: I-MOVE-COVID-19 Study

**DOI:** 10.3390/v16111705

**Published:** 2024-10-30

**Authors:** Miriam Latorre-Millán, María Mar Rodríguez del Águila, Laura Clusa, Clara Mazagatos, Amparo Larrauri, María Amelia Fernández, Antonio Rezusta, Ana María Milagro

**Affiliations:** 1Research Group on Infections Difficult to Diagnose and Treat, Miguel Servet University Hospital, Institute for Health Research Aragón, 50009 Zaragoza, Spain; lclusa@iisaragon.es (L.C.); arezusta@salud.aragon.es (A.R.); amilagro@salud.aragon.es (A.M.M.); 2Servicio de Medicina Preventiva, Hospital Universitario Virgen de las Nieves, 18014 Granada, Spain; mmamelia.fernandez.sspa@juntadeandalucia.es; 3National Centre of Epidemiology, Carlos III Health Institute, 28029 Madrid, Spain; cmazagatos@isciii.es (C.M.); alarrauri@isciii.es (A.L.); 4Consortium for Biomedical Research in Epidemiology and Public Health (CIBERESP), 28029 Madrid, Spain

**Keywords:** COVID-19, SARS-CoV-2, severity, death, ventilation, ICU, clinical phenotype

## Abstract

In the frame of the I-MOVE-COVID-19 project, a cohort of 2050 patients admitted in two Spanish reference hospitals between March 2020 and December 2021 was selected and a range of clinical factor data were collected at admission to assess their impact on the risk COVID-19 severity outcomes through a multivariate adjusted analysis and nomograms. The need for ventilation and intensive care unit (ICU) admission were found to be directly associated with a higher death risk (OR 6.9 and 3.2, respectively). The clinical predictors of death were the need for ventilation and ICU, advanced age, neuromuscular disorders, thrombocytopenia, hypoalbuminemia, dementia, cancer, elevated creatin phosphokinase (CPK), and neutrophilia (OR between 1.8 and 3.5), whilst the presence of vomiting, sore throat, and cough diminished the risk of death (OR 0.5, 0.2, and 0.1, respectively). Admission to ICU was predicted by the need for ventilation, abdominal pain, and elevated lactate dehydrogenase (LDH) (OR 371.0, 3.6, and 2.2, respectively) as risk factors; otherwise, it was prevented by advanced age (OR 0.5). In turn, the need for ventilation was predicted by low oxygen saturation, elevated LDH and CPK, diabetes, neutrophilia, obesity, and elevated GGT (OR between 1.7 and 5.2), whilst it was prevented by hypertension (OR 0.5). These findings could enhance patient management and strategic interventions to combat COVID-19.

## 1. Introduction

COVID-19 is a disease in response to a viral infection, the Severe Acute Respiratory Syndrome Coronavirus 2 (SARS-CoV-2), which emerged in Wuhan, China, in December 2019. The clinical pathway may progress from an asymptomatic state to a more severe illness, including pneumonia, acute respiratory distress syndrome (ARDS), and fatal multi-organ dysfunction. The effects of the disease can be characterised in terms of clinical severity, with reference to the medical complications experienced by infected individuals. In this regard, the disease has been classified in accordance with various definitions, including those set forth by the World Health Organization (WHO) [1].

The present study was conducted in the frame of a multidisciplinary European hospital network for the research, prevention, and control of the COVID-19 pandemic (I-MOVE-COVID-19 Hospital Network) [2]. This consortium comprises numerous partners who were already engaged in research activities within the Influenza Monitoring Vaccine Effectiveness in Europe (I-MOVE) network [3]. They rapidly adapted their infrastructure and protocols to incorporate the surveillance of cases of the novel coronavirus in hospitalised patients in several countries. Spain joined the project with the participation of two hospitals that were able to provide valuable information for COVID-19 surveillance in Europe [4], heading also a pilot model for COVID-19 and severe acute respiratory infections (SARI) surveillance at a national level [5].

During the period encompassed by the present study, two noteworthy variants of concern (VOCs) circulated, each exhibiting distinct severity presentations. They were Alpha/B.1.1.7 in winter 2020 and Delta/B.1.617.2 in spring 2021 (named by Phylogenetic Assignment of Named Global Outbreak (PANGO)/lineage designation) [4]. Despite the effectiveness of the COVID-19 vaccines in reducing the incidence of infections, hospitalisations, and deaths, the burden of clinical management remains significant [6]. Furthermore, the virus continues to evolve, with new variants being monitored by international health authorities [7]. However, the pathogenesis of the disease is primarily determined by host-related factors rather than viral genetic mutations. The severity of a COVID-19 episode depends on the host’s underlying health status, as well as presence of pre-existing risk factors and comorbidities [8,9].

As the clinical presentation of COVID-19 is polymorphic [10], it seemed obvious that patients with a phenotype characterised by a pool of risk factors would have a common number of concomitant features on admission that could predict the severity outcome. COVID-19 patients exhibit a broad clinical spectrum, with the disease affecting numerous organ systems, including the respiratory, neuropsychiatric, and cardiovascular systems, as well as manifestations from other organ systems, such as the endocrine, gastrointestinal, renal, and cutaneous systems [11]. However, a meta-analysis of 5000 patients found that only dyspnoea was associated with severity [12]. Moreover, comorbidities and underlying conditions play a major role in disease prognosis [8,9]. These factors have been associated with severe clinical outcomes [11] and are linked to an increased number of biomarkers [13], underscoring their influence on systemic inflammation and immune response [14]. Furthermore, COVID-19 severity outcomes are influenced by a range of sociodemographic factors, including age, sex, occupation, housing circumstances, and barriers to healthcare access [15].

In this context, the integration of sociodemographic, clinical, comorbidity, and metabolomic data enhances the predictive accuracy of the models, and it may be useful to investigate the correlations with severity outcomes. Yet, to the best of our knowledge, the combination of all these features with the aim of predicting several severity outcomes has been conducted on a small number of American COVID-19 patients [16]. Nevertheless, previous studies have examined epidemiological and clinical characteristics associated with specific severity outcomes [10,17,18,19]. However, there are only a limited number of published studies with large sample sizes describing the phenotype associated with severity outcomes in Spanish hospitalised COVID-19 patients [10,20].

Therefore, the present article aims to describe the phenotype at hospital admission associated with severe COVID-19 disease in a sample of Spanish hospitalised COVID-19 cases, providing a comprehensive assessment of a wide range of health-related features and their potential impact as predictors of severity, using ventilation, ICU admission, and death as the main outcomes.

## 2. Materials and Methods

### 2.1. Study Setting and Design

A retrospective multi-site observational study was conducted in the two tertiary care hospitals participating as the Spanish component of the I-MOVE-COVID-19 network, with data collected between 16 March 2020 and the end of 2021. The Miguel Servet (HUMS) and the Virgen de las Nieves (HUVN) University Hospitals, located in Zaragoza and Granada, respectively, are both reference hospitals for the surveillance of respiratory infections within their respective region, with a capacity of more than 1300 beds each and covering a total catchment area of over 880,000 people.

### 2.2. Participants

Following the I-MOVE-COVID-19 surveillance, risk factors, and vaccine effectiveness studies’ protocols [21,22,23], patients admitted to these hospitals during the study period were tested for SARS-CoV-2 detection. Data were subsequently collected in a structured online questionnaire which included a comprehensive set of demographic, epidemiological, and clinical variables. To manage the considerable workload of data collection in periods of high incidence of COVID-19 admissions (from July 2020 and until the end of the study), a systematic selection process was implemented, whereby a weekly sample of patients were chosen (those admitted on every Tuesday and Wednesday).

This study was conducted on a representative sample of 2050 patients that were admitted to one of the aforementioned hospitals and had a positive test for SARS-CoV-2 detection at admission or during the two previous days and only once during 28 days after their index positive swab (in order to ensure their admission was associated to that COVID-19 episode and prevent multiple recruitments for patients being readmitted with an additional positive test for the same index infection). Patients who tested negative for SARS-CoV-2 PCR on nasopharyngeal swab and those with missing data were excluded from the study.

### 2.3. Laboratory Testing

The diagnosis and confirmation of SARS-CoV-2 infection were based on molecular tests performed on patients’ nasopharyngeal swab specimens collected by trained nurses. Concretely, CE-IVD real-time PCR (real-time reverse transcriptase-polymerase chain reaction) was employed, using at least two specific targets of SARS-CoV-2 RNA and an internal control to ensure the quality of the result. The results were classified according to the manufacturer’s specifications as follows: negative (absence of SARS-CoV2 target detection or detection over 38 Ct and detection of internal control), positive (detection of two SARS-CoV-2 targets under 38 Ct), and inconclusive (only one SARS-CoV-2 target detection). Inconclusive samples were retested for clarification and if uncertainty persisted, a new sample from the participant was analysed.

Other laboratory determinations (shown on tables below) were performed on blood samples collected at the time of admission using standardised and similar methods available in hospitals.

### 2.4. Data Collection

Data were obtained with standardised data collection forms from medical electronic records and subsequently anonymised and linked in a data repository. In accordance with the pandemic emergency exemption and the inclusion of these data as part of the pilot surveillance system of the future integrated respiratory surveillance in Spain, no individual patient consent was required [5]. Nonetheless, only summary data were extracted to minimise the risk of disclosure.

### 2.5. Outcomes of Interest

We described the epidemiological and clinical characteristics in relation to the main severity outcomes as the dependent variables. These were the requirement for ventilation, intensive care unit (ICU) admission, and death in COVID-19 hospitalised patients. A comprehensive list of socio-demographics, underlying diseases, symptoms, and biochemical determinations were considered as the independent variables (shown in tables below). The definition of the variables were outlined in the generic published protocol [23].

### 2.6. Statistical Analysis

First, we carried out a descriptive analysis of each epidemiological and clinical variable by sex. Categorical variables were presented as absolute and relative frequencies (n, %) and continuous variables as mean and standard deviations ($\bar{x}$ ± SD). Participants with >30% records with missing data for each specific analysis were excluded. Variables were excluded from the analysis if they exhibited a missing data rate exceeding 30%. Group-wise comparisons were performed using Student’s *t*-test or Chi-square (χ^2^), and by applying Haldane Ascombe correction when needed.

Bivariate logistic regression models were used to assess the association between clinical variables and the main outcomes; need for ventilation, ICU admission, and in-hospital death, with the OR and confidence interval (95%). Models were refitted with categories for continuous variables such as biochemistry results or clinical measurements at admission according to accepted values for the standard range to obtain odds ratios by categories of low/within range/high values as appropriate.

Variables with statistical significance or near-significance were carried forward to a multivariate logistic regression analysis based on the Akaike information criterion (AIC), assessing the independent association of each specific variable with each outcome. However, vaccination and the admission centre were not included in these models due to potential biases resulting in differences in vaccine availability. The final model was determined by stepwise selection criteria based on improvement in AIC, and its suitability was assessed using the area under the receiver operating characteristic curve (ROC AUC score), Nagelkerke’s R^2^ coefficient, accuracy, specificity, and sensitivity values. Finally, nomograms were constructed for each model to serve as a clinical tool. The score associated with the level of risk (low or high) for each nomogram was calculated based on the frequency of each outcome observed in the sample, and its correspondence with the percentile of the predictive risk value calculated for each participant when applying each multivariate prediction model. To construct the nomogram in such a way as to allow for the calculation of an overall score for a given patient, the partial score graphically assigned to the presence of each predictive factor was represented by a line whose length was calculated according to its relative adjusted effect on the outcome (ORs were visually normalised), as determined by the model analysed, and whose upper point scale coincided with the score to be added or subtracted (depending on whether it was a risk or protective factor) to the overall score, whose scale also coincided graphically in length with the level of risk assigned.

Two-tailed *p* value was considered statistically significant when *p* < 0.05. All analyses and graphs were carried out with the free based R software Jamovi 2.2.5 (The Jamovi Project, Australia), MS Excel 2016 (Microsoft, WA, USA), MS PowerPoint 2016 (Microsoft, WA, USA) and MS Word 2016 (Microsoft, WA, USA).

### 2.7. Ethical Approval

This study was conducted in accordance with the Declaration of Helsinki and its subsequent modifications. The requirement for written informed consent was waived, given the context of emerging infectious disease. Nevertheless, the protocol was approved by the Ethics Committee of Andalucía and by the managements of both centres.

## 3. Results

### 3.1. Descriptive Analysis

The characteristics of the studied sample of 2050 hospitalised COVID-19 patients, overall and stratified by sex, are presented in Table 1.

Regarding overall sociodemographic and clinical factors, there were slightly more men than women (53% vs. 47%), their median age was 64 years, and the most common age range was from 40 to 64 years (40%). HUVN admitted 41% participants, whereas 59% were recruited at HUMS. Their vaccine coverage was 10% for COVID-19 (24% since it was available) and 39% for flu. They account for an average of 2.5 previous hospitalisations and seven medical visits in the year prior to their admission (15% added up to more than 12).

In terms of the main severity outcomes, 15% cases died, 9% were admitted to the ICU, and 27% needed ventilation. On average, it took less than a week from the onset of symptoms to hospitalisation and 10 days to ICU admission. The length of stay was over a week for 60% participants and, on average, the stay was over 12 days at the hospital and almost 24 days at the ICU. The time between admission to hospital and ICU was about 4 days, and less than 8% of participants waited more than a week.

Chronic conditions were present in COVID-19 (almost three, on average), with obesity and hypertension the most prevalent, being in nearly half of the sample (48%), with a BMI of 29.4 kg/m^2^ on average. Other frequently occurring disorders included were rheumatic disease (24%), diabetes (24%), and heart disease (20%).

The most prevalent symptoms were cough (71%), feverish (67%), dyspnoea (66%), and malaise (62%), followed by fever (34%), general deterioration (33%), myalgia (24%), diarrhoea (18%), and headache (16%).

The clinical measures exhibited a low average oxygen saturation (92%) and high respiratory rate (26 rpm on average). Indeed, low oxygen saturation was present in 62% participants, whilst other parameters such as systolic and diastolic blood pressure (SBP and DBP) and long QT were out of range in less than 15%.

Regarding biochemistry alterations, the most frequently observed deviations were high levels of ferritin and C-reactive protein (CRP) (99%), urea (95%), lactate dehydrogenase (LDH) (77%), D-dimer (68%), gamma-glutamyl transferase (GGT) (57%), and aspartate transaminase (AST) (47%), it being also frequent to find high levels of alanine transaminase (ALT) (34%), hypoalbuminemia (28%), creatinine phosphokinase (CPK) (25%), neutrophilia (18%), thrombocytopenia, and high prothrombin time (17%).

Compared to women, men were younger and more likely to need ventilation and ICU admission, to have heart and lung diseases, to show fever and long QT, to show higher figures for diastolic and systolic blood pressure, and to show elevated figures for ALT, AST, total bilirubin, C-reactive protein, GGT, prothrombin time, and blood urea, as well as to present thrombocytopenia. Otherwise, men were less likely to be vaccinated against influenza, to account for medical visits, to have anaemia, asthma, dementia and rheumatic diseases, to show chest pain, coryza, diarrhoea, general deterioration, headache, nausea and vomiting, to show low SBP, DBP, and oxygen saturation, and to show elevated D-dimer, neutrophilia, and thrombocytophilia.

Additionally, the temporal distribution in terms of the month of admission for our COVID-19 hospitalised cases resulting from the recruitment implementation described in Section 2.2 is shown in Figure 1.

Furthermore, their distribution by age and sex is represented in Appendix A, showing there were more women in their 80’s but more men in their 50’s.

### 3.2. Bivariate Analysis

The results of the univariate analysis are presented as Appendix A.

As Appendix A shows, death was more likely among those who were older, previously visited their physician more frequently, required ventilation or admission to the ICU, stayed longer at the hospital, remained longer in the ICU, had a greater number of underlying conditions (including anaemia, cancer, dementia, diabetes, heart disease, hypertension, ictus, kidney disease, neuromuscular disorders, and lung or rheumatic disease), or showed confusion, general deterioration, tachycardia, neutrophilia, thrombocytopenia, hypoalbuminemia, low SBP, DBP, oxygen saturation, elevated AST, bilirubin, CPK, D-dimer, LDH, prothrombin time, and urea. Conversely, death was less likely among those who had received flu vaccination and those who showed ageusia, anosmia, chest pain, cough, diarrhoea, fever, feverish, headache, myalgia, sore throat, and vomiting, as well as lower figures for DBP and oxygen saturation and elevated ALT and GGT.

In addition, Appendix A shows that ICU admission was more likely among those of extreme age groups, who were hospitalised at HUVN, who had more than 12 medical visits last year, died or needed ventilation, whose stay was longer, and who showed higher BMI, obesity, diabetes, showed abdominal pain, dyspnoea, low SBP and oxygen saturation or elevated ALT, AST, bilirubin, CPK, GGT, LDH, urea, and neutrophilia. Otherwise, ICU admission was less frequent among those who were men, COVID-19 vaccinated, or showed confusion, malaise, and a higher oxygen saturation.

Appendix A shows that the need for ventilation was more prevalent among those COVID-19 patients who were older, men, admitted to HUVN, flu vaccinated, had more previous medical visits last year, deceased, stayed at ICU, stayed longer, had higher BMI, cancer, diabetes, heart disease, hypertension, lung disease, and obesity, and showed dyspnoea, low SBP and oxygen saturation, elevated AST, bilirubin, CRP, CPK, D-dimer, GGT, LDH, urea, and neutrophilia. Contrary, the likelihood of requiring ventilation was reduced among those who were COVID-19 vaccinated, those with anaemia, rheumatic disease, ageusia, anosmia, chest pain, coryza, diarrhoea, feverish, malaise, and those with lower SBP, DBP, and oxygen saturation levels.

### 3.3. Multivariate Analysis

The independent associations of several risk factors with the severity outcomes computed by adjusted multivariate analysis are shown as forest plots in Figure 2. The statistics for the predictive models are shown in Table 2, including those for all the dataset and also its application on each hospital.

Death was found to be directly associated with several factors: the need for ventilation (OR 6.9, IC 95% 4.3–11.1), having advanced age (OR 3.5, IC 95% 2.6–4.7), ICU admission (OR 3.2, IC 95% 1.8–5.6), neuromuscular disorders (OR 2.9, IC 95% 1.2–2.9), thrombocytopenia (OR 2.2, IC 95% 1.4–3.6), elevated prothrombin time (OR 2.1, IC 95% 1.2–3.3), hypoalbuminemia (OR 2.1, IC 95% 1.3–3.2), dementia (OR 2.0, IC 95% 1.1–3.6), cancer (OR 1.9, IC 95% 1.0–3.4), elevated CPK (OR 1.9, IC 95% 1.2–2.9), and neutrophilia (OR 1.8, IC 95% 1.1–2.9). Inverse associations with death were also found: having cough (OR 0.5, IC 95% 0.4–0.8), sore throat (OR 0.2, IC 95% 0.1–0.8), and vomiting (OR 0.1, IC 95% 0.0–06). Statistics for this model were AUC ROC = 0.803, Nagelkerke’s R^2^ = 0.294, accuracy = 0.813, specificity = 0.956, and sensitivity = 0.279.

ICU admission was found to be directly associated with the need for ventilation (OR 371, IC 95% 90.8–1516.8), abdominal pain (OR 3.6, IC 95% 1.2–10.8), and elevated LDH (OR 2.2, IC 95% 1.2–10.8). Similarly, ICU admission was inversely associated with advanced age (OR 0.5, IC 95% 0.4–0.7). Statistics for this model were AUC ROC = 0.927, Nagelkerke’s R^2^ = 0.520, accuracy = 0.905, specificity = 0.984, and sensitivity = 0.141.

The need for ventilation was found to be directly associated with low oxygen saturation (OR 5.2, IC 95% 2.8–9.4), elevated LDH (OR 3.0, IC 95% 1.5–6.3), elevated CPK (OR 2.4, IC 95% 1.5–3.7), diabetes (OR 2.3, IC 95% 1.4–3.9), neutrophilia (OR 2.2, IC 95% 1.3–3.7), obesity (OR 1.9, IC 95% 1.2–3.0), and elevated GGT (OR 1.7, IC 95% 1.1–2.7). On the contrary, hypertension was inversely associated with the need for ventilation (OR 0.5, IC 95% 0.3–0.8). Statistics for this model were AUC ROC = 0.906, Nagelkerke’s R^2^ = 0.471, accuracy = 0.881, specificity = 0.967, and sensitivity = 0.352.

Some factors lost their statistical significance when the models were applied to each hospital. For death, these were cough, hypoalbuminemia, elevated CPK, sore throat, and vomiting in HUMS, and cancer, dementia, neutrophilia, elevated prothrombin time, neuromuscular disorders, sore throat, and vomiting in HUVN. For ICU admission, significance was lost for abdominal pain and elevated LDH in HUMS and for elevated LDH and ventilation in HUVN. The analysis revealed that ventilation did not demonstrate statistical significance for elevated GGT, neutrophilia, hypertension, and obesity in HUMS; nor for diabetes, elevated CPK, GGT, LDH, neutrophilia, and obesity in HUVN.

Finally, nomograms (Figure 3) were constructed for each outcome based on the aforementioned results. A score of 67 or higher was considered indicative of a high risk of mortality, with a predictive value of 0.338, corresponding to approximately 70 points. A high risk of ICU admission was set for a score of more than 85 points, corresponding to a model predictive value of more than 0.455. Similarly, a high risk of ventilation was set for a score of more than 58 points, corresponding to a model predictive value of 0.293.

## 4. Discussion

We described the clinical phenotypes at hospital admission associated with three different severity outcomes for confirmed COVID-19 cases: death, ICU admission, and the need for ventilation. A large sample size (*n* = 2050) and a broad range of data collected over nearly two years since the start of the COVID-19 pandemic in Spain enabled us to assess a wide range of clinical parameters to characterise these phenotypes with the worst episode prognosis, including socio-demographic factors, underlying comorbidities, and chronic conditions, as well as symptoms and physiological and biochemical markers during admission. Moreover, the description of the associations between the phenotype characteristics and the severity outcomes allows for the construction of predictive models and nomograms that may be useful as clinical tools for estimating the risk of each severity outcome.

There are a substantial number of studies conducted on each feature and outcome included in our aim individually, but only a limited number encompassed all of them or were made in a sufficiently large sample of Spanish COVID-19 patients. Nevertheless, as our data contributed to an European project, we can compare some assessments with the European dataset results [4]. Regarding main outcomes, the situation of severity for the COVID-19 hospitalised patients appears to have been more favourable in Spain, as evidenced by the lower rates for death (15.1% vs. 18.8%), ICU admission (8.7% vs. 13.7%), and use of ventilation (26.7% vs. 38.4%), even accounting for a similar age and sex distribution on both studies (median age was 64.0 vs. 63.8 years, and 53.1% were men vs. 54.6%). On the contrary, our Spanish figures were considerably higher for pluripathology (two or more underlying conditions were present in 62.2% of Europeans but 70.6% of Spaniards), and the prevalence of obesity (48.4% vs. 27.1%), hypertension (47.8% vs. 43.6%), and rheumatic diseases (24.5% vs. 6.3%) were higher, albeit heart disease (19.8% vs. 32.6%) and diabetes (23.8% vs. 26.1%) were lower. Nevertheless, on the Spanish general population and beyond the I-MOVE study, the metabolic risk factors related with death and an elevation for the risk for transmissible diseases are also included at the top of the list of risk factors for death (hypertension 18.5%, diet 17.4%, smoking 14.1%, excess weight 12%, hyperglycaemia 8.7%, and hyperlipidaemia 6.5%) [24], and smoking is even above the European median in Spain (23.1% vs. 19.7%) [25].

Therefore, additional factors must be involved in these differences in severity outcomes by country, such as the manner in which the pandemic crisis and health intervention strategies were managed. In this regard, it is noteworthy that the Spanish healthcare system is characterised by universal coverage and that prior to the pandemic, it was known as one of the best and most effective in Europe, according to the World Economic Forum and to Bloomberg [26,27], highly trusted by the public opinion [28], and supported by the government, which has a strong role [29]. This resulted in the implementation of early, strong mandatory physical distancing measures and a prioritisation of vaccination policies for the elderly, conditioning implications for patient management. Consequently, our study reflects this situation not only in terms of the severity outcomes, but also in the higher median ICU stay for Spanish COVID-19 hospitalised patients when compared to their European counterparts (23.9 vs. 18 days).

On the other hand, our findings on the predictive risk factors for severity outcomes are consistent with literature [30] and aligned with reports from previous large-scale studies on Spanish COVID-19 hospitalised patients, which described some risk factors strongly associated with death hazard, including a number of underlying conditions and elevated inflammatory parameters [10], such as advanced age, low oxygen saturation, and high C-reactive protein [20].

The final models were good or even greater according with the figures for AUC-ROC, accuracy, and specificity (0.80–0.93, 0.81–0.90, and 0.96–0.98, respectively, although figures for Nagerkerke’s R^2^ and sensitivity were not so good (0.294–0.520 and 0.141–0.352). The dichotomisation of variables simplifies the interpretation and construction of the models and facilitates the identification of such features’ profile.

It should be noted that none of the identified predictors appeared in the final models for all severity outcomes (however, age, elevated CPK, LDH, and neutrophilia appeared in two of them). Additionally, several of them were found to be associated in a different (advanced age and hypertension) or not expected direction (vomiting, sore throat, and cough for death). In this regard, we should point out some issues.

Advanced age may be directly associated with death and inversely with admission to the ICU, as this condition does not benefit the patient when applying admission triage in a situation of work overload [31]. Further, hypertension has been identified as a protective factor for the need for ventilation. In this sense, it should be highlighted that SARS-CoV-2 uses angiotensin-converting enzyme 2 (ACE2) as a cell receptor for viral entry, whereas ACE is the target for certain therapeutic treatments for hypertension that could prevent vasoconstriction and disease conditions in Raynaud’s phenomenon. Consequently, patients treated for hypertension may exhibit a distinctive immune response to SARS-CoV-2 which could potentially influence the severity of the disease and thus the necessity for respiratory support [32]. In addition, it is noteworthy that the reporting of some symptoms (such as vomiting, sore throat, and cough) may serve as indicators of disease manifestation, facilitating early diagnosis and treatment [33], fitting with the timeline of symptom presentation previously reported by other authors [34].

It is also important to note that none of the final models incorporated sex, medical contacts (either admissions or visits), or time between events (such as days of stay) as predictors. This can be readily explained by the fact that it is not these factors in isolation, but rather some associated conditions also present in the models, that are playing a role in predicting the severity of COVID-19. However, these factors are easily discernible, and their consideration may be helpful in practice.

Further, some factors deserve additional regard, although their exclusion from the final predictive models was necessary due to our sample characteristics in order to avoid different forms of bias. COVID-19 and flu vaccination has been shown to be effective in reducing severity outcomes, which is evidenced by the inverse associations with death (for flu vaccination), need of ventilation, and ICU admission (for COVID-19 vaccination). However, flu vaccination was found to be directly associated with the need for ventilation; this may be explained by the increased efforts of vaccination campaigns in those at higher risk (e.g., COPD patients), as well as by a very low vaccination coverage. Nonetheless, the lack of COVID-19 vaccine effectiveness for death, or even its perceived risk for the need for ventilation, may be biased in our participants by the timeliness of vaccination availability, as previously mentioned.

Finally, models did not fit similarly when applied to each hospital’s data, showing several statistical differences. This was to be expected, given the notable discrepancies between the hospital datasets, such as the higher number of participants admitted to the ICU and requiring ventilation at HUVN. Nevertheless, the combination of both datasets enabled the construction of a more robust and useful predictive model for other Spanish hospitals. It should be noted that severity outcomes are not solely dependent on differences in SARS-CoV-2 incidence and strains, but also on the characteristics of the catchment population and the availability of resources at each healthcare centre.

### Limitations and Strengths

Our study has limitations that may introduce some bias and that are unavoidable in a pandemic context, when the timeliness of the surveillance system is compromised by other priorities and challenges, such as the considerable work overload in hospitals. Firstly, it was not feasible to include all consecutive patients in the study; therefore, a systematic sample was used prevent introducing additional bias. Similarly, the number of deaths attributed to COVID-19 may be overestimated as the primary outcome was death due to any cause during the index hospital admission, irrespective of the recorded cause of death.

On the other hand, the data collection methods affected the data completeness of certain variables, resulting in missing data. Even though missing data were excluded solely only for the analysis of each variable (to prevent loss of power), this may have impacted the accuracy of results and their generalisability. Poor reporting from the health care workers when they were overloaded with work, particularly in the initial stages of the pandemic, may also have contributed to this type of bias. Additionally, COVID-19 vaccination coverage data may be biased by vaccine availability and the complexity of calculation, as the population was vaccinated in a staggered manner according to age and other priority conditions.

In this sense, the lower predictive ability for the need for ventilation model may be attributed to the absence of a detailed characterisation of the ventilation type required, ranging from patients who used nasal goggles to those with invasive mechanical ventilation. However, including only those patients with mechanical ventilation would not have substantially improved our results either, as the sample size would be smaller and thus also the power and reliability of our findings. Likewise, our models have low sensitivity and are therefore biased in correctly identifying people who will ultimately have a dismal outcome. With this in mind, it should be noted that the consideration of other previously reported strongly associated factors beyond those investigated in this paper, such as the assessment of chest X-ray infiltrates [35], could help to increase the sensitivity and predictive ability of these models.

Notwithstanding the aforementioned challenges, the design of the definitions of the variables studied and the data collection were carried out according to a published robust protocol from a European project of excellence (H2020). A wide list of clinical sociodemographic and clinical variables was assessed on a large sample size. These findings have implications for the high accuracy revealed by the high AUC scores obtained in the multivariate analysis. Moreover, the study was conducted in two tertiary referral hospitals situated in different Spanish regions; thereby conferring representativeness at the national level.

Additionally, to the best of our knowledge, this is the first study describing predictive models and nomogram tools for the phenotype association with death, ICU stay, and the need of ventilation, simultaneously, in Spanish COVID-19 hospitalised patients and through a comprehensive range of factors at admission, including underlying conditions, as well as symptoms and biochemical parameters.

Furthermore, this study contributed to the European I-MOVE-COVID-19 surveillance system and laid the groundwork for launching the COVID-19 Spanish official surveillance system. Likewise, it emphasises the necessity of monitoring hospitalised patients to understand phenotypes features that may predict severe outcomes, thereby informing public health responses, particularly in the context of emerging diseases.

## 5. Conclusions

This study assessed the clinical phenotype of Spanish COVID-19 hospitalised patients in relation to three main severity outcomes by computing predictive models, which in turn have enabled the development of nomograms as clinical tools for prognosis.

Death, ICU admission, and the need for ventilation were found to be affected by the role of several risk factors (advanced age; diagnosis of obesity, cancer, diabetes and neuromuscular disorders; presence of abdominal pain and cough; low blood levels of oxygen saturation, platelets and albumin; and elevated blood levels of CPK, GGT, LDH, neutrophils, and elevated prothrombin time), as well as protective factors (diagnosis of hypertension and the presence of cough, sore throat, and vomiting). Predictive models including these features allowed the construction of nomograms as clinical tools to assess the severity risk of illness for different outcomes. Furthermore, additional useful associations have been identified, including vaccination status, the number of underlying conditions (diseases and previous medical visits or admissions), and characteristics of the stay (such as the time between events or the centre where admitted).

These findings should be considered in order to improve COVID-19 patient management and the implementation of strategic interventions, including policy measures to prevent and combat COVID-19, the allocation of available resources in a more efficient way, and the avoidance of overstressing the healthcare system. Studies with similar approaches may be useful in addressing these issues in the context of future emerging diseases.

## Figures and Tables

**Figure 1 viruses-16-01705-f001:**
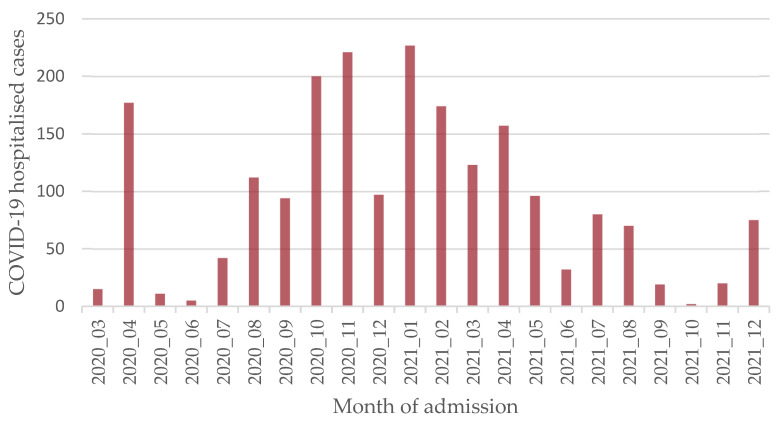
Spanish COVID-19 hospitalised cases by month (I-MOVE-COVID19 study).

**Figure 2 viruses-16-01705-f002:**
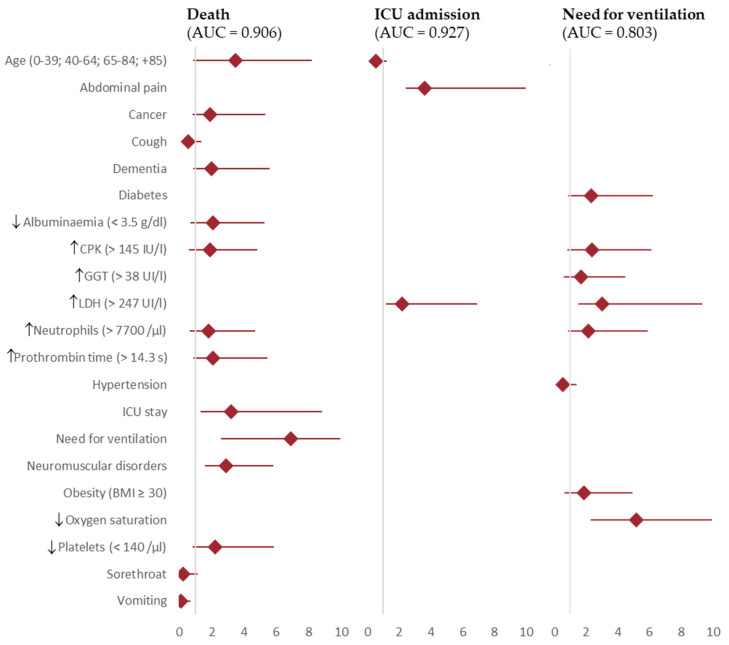
Predictive factors associated with ventilation, ICU admission, and death in Spanish hospitalised patients. I-MOVE-COVID19 study. Note ↓ and ↑ indicates that the predictive factor is over or under the normal range respectively.

**Figure 3 viruses-16-01705-f003:**
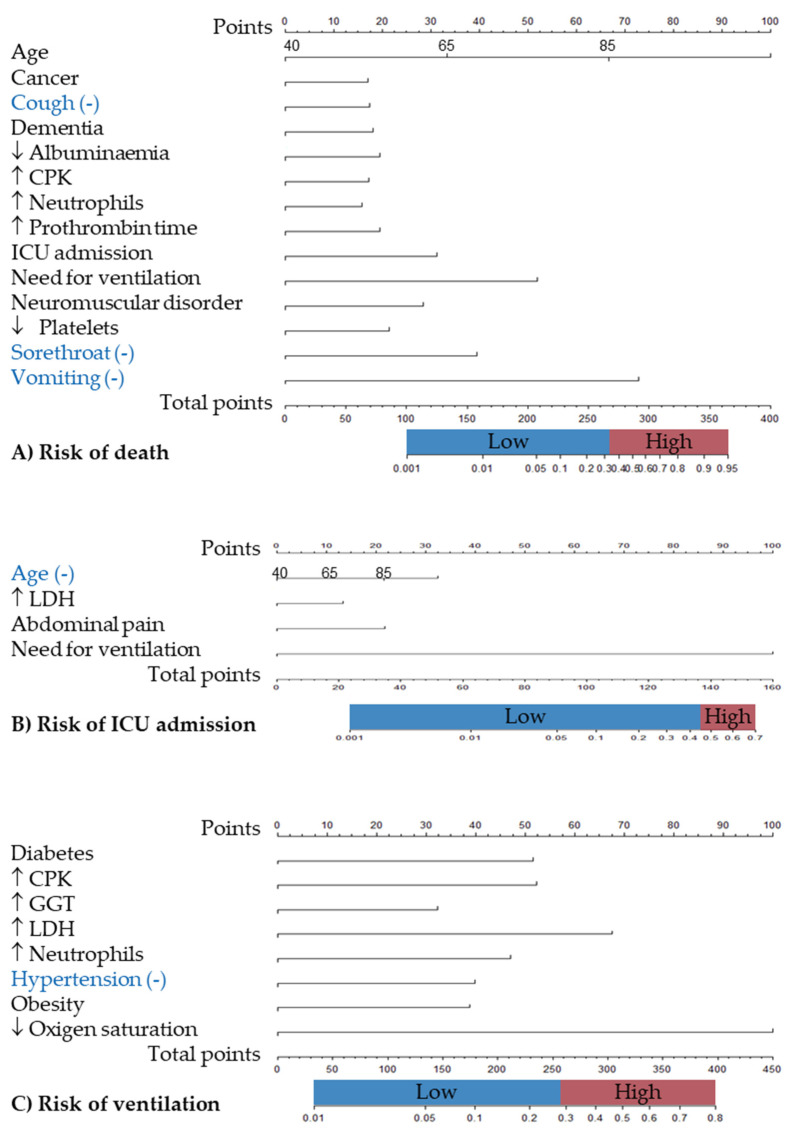
Nomograms for predicting COVID-19 severity outcomes in Spanish hospitalised patients. I-MOVE-COVID19 study. (**A**) Death, (**B**) ICU admission, and (**C**) need for ventilation. Note ↓ and ↑ indicates that the predictive factor is over or under the normal range respectively.

**Table 1 viruses-16-01705-t001:** Epidemiological and clinical characteristics of Spanish COVID-19 hospitalised patients, by sex (I-MOVE-COVID19 study).

	All 2050 (100%)	Women 961 (46.8%)	Men 1089 (53.1%)	*p* Value
**Sociodemographic and clinical factors**				
Age, grouped (0–39; 40–64; 65–84; 85+ years)				<0.001
Age 0–39 years	215 (10.5)	106 (11.0)	109 (10.0)	
Age 40–64 years	810 (39.5)	332 (34.5)	478 (43.9)	
Age 65–84 years	723 (35.3)	359 (37.4)	364 (33.5)	
Age 85+ years	301 (14.7)	164 (17.1)	137 (12.6)	
Centre (HUVN)	833 (40.6)	374 (38.9)	459 (42.1)	0.137
COVID-19 vaccinated (only when available)	215 (23.6)	91 (22.1)	124 (24.8)	0.345
Flu vaccinated	794 (39.1)	394 (41.4)	400 (37.1)	0.050
Number of previous hospitalisations (last year), mean ± SD	2.5 ± 3.2	2.4 ± 2.1	2.7 ± 3.8	0.653
Number of previous medical visits (last year), mean ± SD	6.8 ± 7.1	7.7 ± 7.4	6.1 ± 6.9	0.009
Previous medical visits last year > 12	89 (15.2)	48 (18.4)	41 (12.6)	0.053
**Outcomes**				
Death	309 (15.1)	138 (14.4)	171 (15.7)	0.402
ICU stay	179 (8.8)	70 (39.1)	109 (60.9)	0.031
Need for ventilation	548 (27.4)	223 (23.7)	325 (30.5)	<0.001
**Time between events**				
Days from onset to admission, mean ± SD	6.6 ± 12.4	6.2 ± 12.7	6.9 ± 12.1	0.166
Days from onset to ICU, mean ± SD	9.9 ± 4.6	9.7 ± 5.1	9.9 ± 4.2	0.750
Days of stay, mean ± SD	12.2 ± 14.6	11.9 ± 16.4	12.6 ± 12.7	0.263
Days of stay (>7 days)	1248 (60.9)	569 (59.7)	679 (62.8)	0.159
Days of ICU stay, mean ± SD	23.9 ± 20.2	23.9 ± 21.9	24.0 ± 19.2	0.969
Days from admission to ICU, mean ± SD	3.8 ± 3.3	3.8 ± 3.8	3.8 ± 2.9	0.881
Days from admission to ICU > 7	14 (7.8)	6 (8.6)	8 (7.3)	0.765
**Underlying conditions**				
Anaemia	241 (11.8)	138 (14.4)	103 (9.5)	<0.001
Asthma	171 (8.4)	117 (12.2)	54 (5.0)	<0.001
Body mass index (BMI), mean ± SD	29.4 ± 5.7	29.4 ± 6.1	29.4 ± 5.3	0.831
Cancer	161 (7.9)	70 (7.3)	91 (8.4)	0.376
Dementia	197 (9.6)	111 (11.6)	86 (7.9)	0.005
Diabetes	487 (23.8)	220 (22.9)	267 (24.5)	0.396
Heart disease	405 (19.8)	161 (16.8)	244 (22.4)	0.002
Hypertension	978 (47.8)	461 (48.1)	517 (47.6)	0.818
Ictus	124 (6.1)	50 (5.2)	74 (6.8)	0.134
Immunodeficiency	39 (1.9)	16 (1.7)	23 (2.1)	0.464
Kidney disease	211 (10.3)	103 (10.8)	108 (9.9)	0.550
Lung disease	216 (10.6)	69 (7.2)	147 (13.5)	<0.001
Liver disease	87 (4.3)	33 (3.4)	54 (5.0)	0.089
Neuromuscular disorders	110 (5.4)	50 (5.2)	60 (5.5)	0.767
Obesity	548 (48.4)	287 (50.7)	261 (46.0)	0.115
Rheumatic disease	501 (24.5)	290 (30.2)	211 (19.4)	<0.001
Multiple pathology (≥2 underlying conditions)	796 (70.6)	400 (70.9)	396 (70.2)	0.794
**Symptoms**				
Abdominal pain	59 (2.9)	34 (3.6)	25 (2.3)	0.098
Ageusia	159 (7.9)	86 (9.0)	73 (6.8)	0.061
Anosmia	169 (8.4)	88 (9.3)	81 (7.5)	0.061
Chest pain	263 (12.9)	147 (15.4)	116 (10.8)	0.002
Chills	67 (3.3)	30 (3.2)	37 (3.5)	0.706
Confusion	76 (3.7)	37 (3.9)	39 (3.6)	0.774
Coryza	67 (3.3)	46 (4.8)	21 (2.0)	<0.001
Cough	1435 (70.6)	685 (71.9)	750 (69.5)	0.242
Diarrhoea	368 (18.1)	196 (20.5)	172 (16.0)	0.008
Dizzy	92 (4.5)	47 (4.9)	45 (4.2)	0.426
Dyspnoea	1334 (65.6)	629 (65.9)	705 (65.3)	0.756
Fever	680 (33.5)	286 (30.0)	394 (36.6)	0.002
Feverish	1365 (67.1)	637 (66.6)	728 (67.5)	0.666
General deterioration	671 (33.1)	346 (36.2)	325 (30.2)	0.004
Headache	327 (16.1)	171 (17.9)	156 (14.5)	0.036
Malaise	1249 (61.5)	588 (61.6)	661 (61.4)	0.928
Myalgia	481 (23.7)	236 (24.7)	245 (22.8)	0.298
Nausea	159 (7.8)	97 (10.2)	62 (5.8)	<0.001
Sore throat	102 (5.0)	53 (5.6)	49 (4.6)	0.307
Tachycardia	572 (28.1)	257 (26.8)	315 (29.2)	0.225
Vomiting	130 (6.4)	83 (8.7)	47 (4.4)	<0.001
**Clinical measures**				
SBP (mm Hg), mean ± SD	129 ± 21.2	126.5 ± 21.6	131.1 ± 20.5	<0.001
SBP < 90 mm Hg	42 (2.2)	29 (3.3)	13 (1.3)	0.004
DBP (mm Hg), mean ± SD	74.4 ± 13.4	73.4 ± 13.7	75.3 ± 13.1	0.002
DBP < 60 mm Hg	228 (12.2)	123 (13.9)	105 (10.7)	0.033
Heart rate (bpm), mean ± SD	91 ± 21.2	91.1 ± 21.0	90.8 ± 21.4	0.728
Long QT on the ECG	50 (6.5)	16 (4.3)	34 (8.5)	0.020
Oxygen saturation (%), mean ± SD	92.2 ± 5.7	92.4 ± 5.7	92.1 ± 5.7	0.174
Low oxygen saturation	1134 (62.2)	500 (58.8)	634 (65.2)	0.005
Respiratory rate (rpm), mean ± SD	25.8 ± 26.6	27.9 ± 37.2	23.8 ± 8.1	0.142
**Biochemical alterations**				
Hypoalbuminemia (albumin < 3.5 g/dL)	469 (28.1)	230 (29.9)	239 (26.6)	0.128
ALT > 35 UI/L	676 (33.7)	259 (27.3)	417 (39.3)	<0.001
AST > 35UI/L	917 (46.9)	384 (41.7)	533 (51.5)	<0.001
Hyperbilirubinemia > 1.2 mg/dL	87 (4.4)	16 (1.7)	71 (6.8)	<0.001
C-reactive protein > 1 mg/dL	1984 (98.6)	931 (98.5)	1053 (98.7)	0.746
CPK > 145 IU/L	366 (25.5)	115 (17.9)	251 (31.6)	<0.001
D-dimer (>500 μg/mL)	1327 (68.0)	649 (71.2)	678 (65.2)	0.005
Eosinophilia (>500 eosinophils/μL)	92 (4.6)	47 (5.0)	45 (4.2)	0.420
Ferritin > 200 ng/mL	1970 (99.6)	922 (99.4)	1048 (99.8)	0.111
GGT > 38 UI/L	1126 (57.0)	467 (50.5)	659 (62.8)	<0.001
LDH > 247 UI/L	1462 (77.5)	679 (77.0)	783 (78.0)	0.602
Neutrophilopenia (<1500 neutrophils/μL)	40 (2.0)	21 (2.2)	19 (1.8)	0.485
Neutrophilia (>7700 neutrophils/μL)	374 (18.4)	157 (16.5)	217 (20.2)	0.030
Thrombocytopenia (<140 platelets/μL)	350 (17.3)	123 (12.9)	227 (21.2)	<0.001
Thrombocytophilia (>370 platelets/μL)	115 (5.7)	66 (6.9)	49 (4.6)	0.022
Prothrombin time > 14.3 s	335 (17.1)	123 (13.4)	212 (20.3)	<0.001
Urea in blood > 20 mg/mL	1902 (94.9)	858 (91.6)	1044 (97.8)	<0.001

Data are expressed by absolute (n) and relative (%) frequencies for categorical variables and by mean ($\bar{x}$) ± standard deviation (SD) for quantitative variables. Figures in bold point for significant differences (*p* value ≤ 0.05). Abbreviations: HUVN: Virgen de las Nieves University Hospital, ICU: intensive care unit, SBP: systolic blood pressure, DBP: diastolic blood pressure, ECG: electrocardiogram, ALT: alanine transaminase, AST: aspartate transaminase, CPK: creatinine phosphokinase, GGT: gamma-glutamyl transferase, LDH: lactate dehydrogenase. The reference category for age is 0–40 years.

**Table 2 viruses-16-01705-t002:** Predictive factors for ventilation, ICU admission, and death in Spanish hospitalised patients (I-MOVE-COVID19 study).

	Death Adjusted OR (IC 95%)			ICU Admission Adjusted OR (IC 95%)			**Need for Ventilation** **Adjusted OR (IC 95%)**		
	All	HUMS	HUVN	All	HUMS	HUVN	**All**	**HUMS**	**HUVN**
Age (<40; 40–64; 65–84; +85 years)	3.5 (2.6–4.7) ***	4.2 (2.4–7.4) ***	3.1 (2.1–4.6) ***	0.5 (0.4–0.7) ***	0.6 (0.3–0.9) *	0.5 (0.4–0.7) ***			
Abdominal pain				3.6 (1.2–10.8) *	3.2 (0.4–28.2)	3.8 (1.1–13.6) *			
Cancer	1.9 (1.04–3.4) *	3.0 (1.1–8.5) *	1.7 (0.8–3.8)						
Cough	0.5 (0.4–0.8) **	0.6 (0.3–1.2)	0.6 (0.3–1.0) *						
Dementia	2.0 (1.1–3.6) *	5.3 (2.1–13.5) ***	0.6 (0.2–1.6)						
Diabetes							2.3 (1.4–3.9) ***	2.5 (1.4–4.6) **	1.9 (0.6–5.6)
Hypoalbuminemia (<3.5 g/dL)	2.1 (1.3–3.2) ***	1.2 (0.6–2.3)	3.9 (2.0–7.9) ***						
Elevated CPK (>145 IU/L)	1.9 (1.2–2.9) **	1.7 (0.8–3.4)	2.1 (1.2–3.8) *				2.4 (1.5–3.7) ***	2.2 (1.3–3.9) **	2.4 (0.9–6.6)
Elevated GGT (>38 UI/L)							1.7 (1.1–2.7) *	1.3 (0.8–2.3)	2.8 (0.9–8.8)
Elevated LDH (>247 UI/L)				2.2 (1.0–4.7) *	2.5 (0.4–28.2)	1.9 (0.7–5.6)	3.0 (1.5–6.3) **	3.2 (1.4–7.5) **	0.8 (0.1–5.5)
Elevated neutrophils (>7700/μL)	1.8 (1.1–2.9) *	4.4 (2.0–9.4) ***	1.3 (0.7–2.4)				2.2 (1.3–3.7) **	1.8 (0.9–3.4)	2.4 (0.7–8.3)
Elevated prothrombin time (>14.3s)	2.1 (1.2–3.3) **	3.4 (1.6–7.1) ***	1.2 (0.6–2.4)						
Hypertension							0.5 (0.3–0.8) **	0.7 (0.4–1.2)	0.3 (0.1–0.8) *
ICU stay	3.2 (1.8–5.6) ***	4.0 (1.2–12.8) *	2.5 (1.3–4.9) **						
Need for ventilation	6.9 (4.3–11.1) ***	11.5 (4.7–28.2) ***	9.9 (4.5–21.7) ***	371 (90.8–1516.8) ***	213.4 (50.3–906.3) ***	5.6 × 10^−8^ (0-inf)			
Neuromuscular disorders	2.9 (1.2–2.9) **	6.9 (2.4–19.5) ***	0.5 (0.1–2.4)						
Obesity (BMI ≥ 30)							1.9 (1.2–3.0) **	1.2 (0.7–2.2)	0.5 (0.1–3.3)
Low oxygen saturation							5.2 (2.8–9.4) ***	4.9 (2.4–10.0) ***	4.9 (1.3–19.3) *
Low platelets (< 140/μL)	2.2 (1.4–3.6) ***	2.2 (1.0–4.6) *	2.7 (1.4–5.3) **						
Sore throat	0.2 (0.1–0.8) *	1.4 × 10^−8^ (0.0–inf)	0.5 (0.1–2.0)						
Vomiting	0.1 (0.0–0.6) *	2.9 × 10^−8^ (0.0–inf)	0.1 (0.0–1.0)						
AUROC	0.906	0.939	0.896	0.927	0.944	0.887	0.803	0.789	0.755
Nagelkerke’s R^2^	0.471	0.571	0.462	0.520	0.522	0.480	0.294	0.231	0.266
Accuracy	0.881	0.918	0.873	0.905	0.950	0.837	0.813	0.855	0.663
Specificity	0.967	0.973	0.961	0.984	0.996	0.963	0.956	0.989	0.605
Sensitivity	0.352	0.532	0.388	0.141	0.073	0.165	0.279	0.909	0.712

Abbreviations: CPK: creatinine phosphokinase, GGT: gamma-glutamyl transferase, LDH: lactate dehydrogenase, s: seconds, ICU: intensive care unit, BMI: body mass index. *p* value: *** ≤0.001; ** ≤0.01; * ≤0.05. The reference for age category is 0–40 years. All data were used for the multivariate model adjustment.

## Data Availability

Due to data protection policies in the hospitals, the complete dataset used for this study is not publicly available, but data can be accessed upon request to the corresponding author.

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
