# Peer review of "Severity Patterns in COVID-19 Hospitalised Patients in Spain: I-MOVE-COVID-19 Study"

_viruses, 2024, doi:10.3390/v16111705_

Round 1

Reviewer 1 Report

Comments and Suggestions for Authors

Dear Authors,

Thank you for your manuscript. Please see my comments below.

The paper is well written and the study is well organised and presents important findings. My comments are mainly technical issues for the authors to consider.

Please explain the abbreviations when they first appear in the abstract (ICU, LDH, CPK).

Line 144: What are C categorical variables?

Table 1, line "mortality" - shouldn't it be replaced with "deaths" or "case fatality" as it is presented as % of deaths of hospitalised COVID-19 patients? Please revise the term throughout the text. Clinical measures section: could "Long QT" be better explained to the reader as "Long QT on the ECG"?

Sorry if I missed this, but it is unclear how the scores in Figure 3 were calculated and how the risk of death, ICU admission and ventilation was determined.

As there are many clinical abbreviations in the paper, could they be listed separately at the end of the text?

Comments on the Quality of English Language

Minor English editing is required.

Author Response

Please explain the abbreviations when they first appear in the abstract (ICU, LDH, CPK). As there are many clinical abbreviations in the paper, could they be listed separately at the end of the text?

Thank you for pointing this out. All abbreviations have been checked to ensure that they are explained by the first time they appear in the abstract and text. We have also included a list of all abbreviations at the end of the manuscript, as suggested in your last point.

Line 144: What are C categorical variables?

Line 144. "C categorical variables" was a typo. It has been replaced with "Categorical variables".

Table 1, line "mortality" - shouldn't it be replaced with "deaths" or "case fatality" as it is presented as % of deaths of hospitalised COVID-19 patients? Please revise the term throughout the text. Clinical measures section: could "Long QT" be better explained to the reader as "Long QT on the ECG"?

Thank you for these suggestions. The term "mortality" has been replaced by "deaths" in Table 1, and has also been checked and replaced throughout the text where appropriate. The term "Long QT" is now first introduced as "Long QT on the electrocardiogram (ECG)" in the results section for clinical measures, and included as "Long QT on the ECG" in all tables. "ECG" has also been added to the list of abbreviations.

Sorry if I missed this, but it is unclear how the scores in Figure 3 were calculated and how the risk of death, ICU admission and ventilation was determined.

We have clarified the explanation of how the level of the risk of our main outcomes was calculated, and how the scores for each factor and for the global risk were calculated. We have also added an explanation of how it works. It is near the end of the methods section.

Minor English editing

We have changed several expressions such as “need for ventilation” instead of “need of ventilation”, or British instead of American spelling (“s” instead of “z” in hospitalised, characterise), etc. We highlighted all changes in the text in yellow

Reviewer 2 Report

Comments and Suggestions for Authors

I reviewed the viruses-3237292 manuscript. The authors evaluated a large cohort of COVID-19 patients, who were hospitalized in two tertiary hospitals in Spain during a 3-year period. The authors tried to identify predictors of dismal outcomes using multiple logistic regression models. The study is well-conducted and adequately presented. However, certain drawbacks are present. The most important ones are the following:

1) The study lacks novelty.

2) The multivariate models exhibit very low sensitivity.

3) The authors did not include consecutive patients and this should be clearly pointed out in the limitations paragraph.

4) The authors should explain the possible impact of recruiting patients with positive PCR test 28-days prior to their admission on the study findings.

5) The authors should clearly define whether all patients in the ventilation group were mechanically ventilated. More specifically it is stated that "our analysis did not include specification of the type of ventilation (i.e. whether ventilation was mechanical or not)" and this needs further clarification.

6) In the limitations paragraph the authors should state that previously reported factors strongly associated with the development of respiratory failure and the need for mechanical ventilation and mortality, such as the extent of chest X-ray infiltrates were not evaluated in this study. In this regard it would be useful to add the following reference: Pappas AG, Panagopoulos A, Rodopoulou A, Alexandrou M, Chaliasou AL, Skianis K, Kranidioti E, Chaini E, Papanikolaou I, Kalomenidis I. Moderate COVID-19: Clinical Trajectories and Predictors of Progression and Outcomes. J Pers Med. 2022 Sep 8;12(9):1472. doi: 10.3390/jpm12091472.

7) The introduction and the discussion are very large and they need shortening, so as to be more reader-friendly.

Comments on the Quality of English Language

Minor English modifications are required.

Author Response

1) The study lacks novelty.

Regarding the lack of novelty, we would have liked to publish this paper earlier, but it was delayed due to the high workload and urgency that the pandemic demanded on other tasks for our patient care commitment and the implementation of the official COVID-19 surveillance system, both necessary for public health decision-making in our country. However, this explanation, which we bring to the attention of the reviewers and editors, may not be of sufficient interest to the readers to be included as such in the article. We agree that other studies published during the pandemic have already addressed COVID-19 severity patterns. Nevertheless, we conducted a comprehensive data collection in a large cohort of hospitalised patients from two different tertiary hospitals, and we believe that our work still provides relevant results that are valuable to share with the scientific community. To our knowledge, there are no published articles in our specific Spanish population that match the contributions of our study, which were already highlighted as strengths in the first version of the manuscript.

2) The multivariate models exhibit very low sensitivity.

The low sensitivity of the models has been included in the limitations section and fits well with the idea of including the lack of use of chest-X-rays and the use of the reference you already provided as an example of how this could be improved. We have also clarified the effect of including a wide range of ventilation types in the ventilation group, as a limitation in this sense.

3) The authors did not include consecutive patients and this should be clearly pointed out in the limitations paragraph.

We add some comments to the limitations section regarding the potential bias related to the inclusion of non-consecutive patients in the first paragraph of the limitations section related to the idea expressed in the first point above. However, to minimise potential bias associated with this, we selected a systematic sample as described in the methods section.

4) The authors should explain the possible impact of recruiting patients with positive PCR test 28-days prior to their admission on the study findings.

Well, we indicated in the methods section as an eligibility criterion that “participants were admitted two days prior and up to 28 days after their index positive swab”, but we did not mean that they could be recruited up to 28 days before to their admission. We have clarified in the methods section that this 28-day criterion is intended to avoid multiple recruitment of a participant if he/she was admitted more than once for the same index infection within a 28-day period. Thus, this 28-day period helps to distinguish between different possible infections, specially at beginning of the pandemic, when the infection period was longer. Thank you for pointing out that this could be misunderstood.

5) The authors should clearly define whether all patients in the ventilation group were mechanically ventilated. More specifically it is stated that "our analysis did not include specification of the type of ventilation (i.e. whether ventilation was mechanical or not)" and this needs further clarification.

We have clarified the wide range of ventilation types used by patients in the ventilation group, and included the lack of specification of ventilation type as a limitation.

6) In the limitations paragraph the authors should state that previously reported factors strongly associated with the development of respiratory failure and the need for mechanical ventilation and mortality, such as the extent of chest X-ray infiltrates were not evaluated in this study. In this regard it would be useful to add the following reference (doi: 10.3390/jpm12091472).

Thank you for the suggestion. We have included the reference in the discussion, and we added some clarification in this text: “… With this in mind, it should be noted that consideration of other previously reported strongly associated factors beyond those investigated in this paper, such as the assessment of chest X-ray infiltrates ([35]), could help to increase the sensitivity and predictive ability of these models.

7) The introduction and the discussion are very large and they need shortening, so as to be more reader-friendly.

The introduction and discussion sections have been shortened by cutting some sentences and paraphrasing others, but only a little, as too much shortening may result in the loss of some potentially valuable information. On the other hand, we have also changed some expressions to improve the reader’s experience.